# Flexible information routing in neural populations through stochastic comodulation

**Caroline Haimerl**
Center for Neural Science
New York University
ch2880@nyu.edu

**Cristina Savin**
Center for Neural Science
Center for Data Science
New York University
csavin@nyu.edu

**Eero P. Simoncelli**
Center for Neural Science, and
Howard Hughes Medical Institute
New York University
eero.simoncelli@nyu.edu

## Abstract

Humans and animals are capable of flexibly switching between a multitude of tasks, each requiring rapid, sensory-informed decision making. Incoming stimuli are processed by a hierarchy of neural circuits consisting of millions of neurons with diverse feature selectivity. At any given moment, only a small subset of these carry task-relevant information. In principle, downstream processing stages could identify the relevant neurons through supervised learning, but this would require many training trials. Such extensive learning periods are inconsistent with the observed flexibility of humans or animals, both of whom can adjust to changes in task parameters or structure almost immediately. Here, we propose a novel solution based on functionally-targeted stochastic modulation. It has been observed that trial-to-trial neural activity is modulated by a shared, low-dimensional, stochastic signal that introduces task-irrelevant noise. Counter-intuitively, this noise appears to be preferentially targeted towards task-informative neurons, corrupting the encoded signal. We hypothesize that this modulation offers a solution to the identification problem, labeling task-informative neurons so as to facilitate decoding. We simulate an encoding population of spiking neurons whose rates are modulated by a shared stochastic signal, and show that a linear decoder with readout weights estimated from neuron-specific modulation strength can achieve near-optimal accuracy. Such a decoder allows fast and flexible task-dependent information routing without relying on hardwired knowledge of the task-informative neurons (as in maximum likelihood) or unrealistically many supervised training trials (as in regression).

## 1 Introduction

Our survival depends on the actions we take, which are derived from internal states and sensory input. Accurate decisions require reliable encoding and flexible task-specific decoding of sensory information. Take for instance the perceptual task of detecting a change in orientation of a grating within a small aperture, placed at a particular location in the visual field (Fig. 1). Neurons in primary visual cortex (V1) that respond selectively to features at different spatial locations and orientations encode the visual stimulus. However, only a small fraction of those neurons would show a change in response when the grating changes orientation (Fig. 1, red); the overwhelming majority will not respond at all or their responses would not change significantly (Fig. 1, gray). Since nearly all visual information passes through V1, any downstream areas' sole source of information is contained in the responses of those few V1 cells. Thus, solving this task relies on the ability to properly gather and combine the responses of these task-relevant neurons, while ignoring the background chatter of activity emanating from the remainder of the population. Furthermore, if the task changes (e.g., due to a change in stimulus position or orientation), the informative sub-population within V1 will

change, and downstream areas will need to modify their processing accordingly. The means by which the brain can achieve such dynamic task-dependent routing of information is a mystery.

The readout of sensory information in neural responses is often explored using statistically optimal decoders derived from specific encoding models. While these decoders can provide an upper bound on performance [1, 2, 3, 4, 5, 6, 7, 8], they should not be interpreted as models for biological decoding, since they generally rely on full knowledge of the stimulus response and noise properties of neurons. It seems inconceivable that upstream decoding circuits could have access to, or store, such detailed information. An alternative possibility is that the decoder is learned from experience. This requires extensive training on the discrimination task, accompanied by feedback regarding the success or failure on each trial. The need for many trials, with feedback, seems inconsistent with the observed behavioral flexibility of animals or humans, both of whom can rapidly adjust to changes in task conditions [9].

Here we propose a novel framework for biologically plausible, flexible decoding, inspired by recent results on task-dependent noise properties of neural populations in the visual system. Neural noise limits the amount of stimulus information that a neural population can encode [10, 1] and is commonly modeled with a Poisson process. However, neurons seem to share sources of multiplicative trial-to-trial variability, or correlated noise, suggesting that additional time-varying modulators influence the response of neurons [11]. Theoretical work indicates that such correlated noise can be detrimental for population encoding, as it cannot be averaged out [7, 12]. Importantly, in some experiments this noise seems to be specifically targeted to neurons that are informative for the task, which further exacerbates the detrimental effects on encoding. Specifically, V4 neurons have been shown to share a common source of noise-modulation, which affects neurons that are informative for the task more strongly [13]. Similarly, V1 noise correlation structure is better explained by task-informativeness than by stimulus tuning properties, suggesting that the source of these correlations is top-down (as opposed to stimulus-driven) [14].

The mechanisms underlying this modulation remain unclear but the observed task-specific structure has functional implications. From an encoding perspective, it is counterintuitive that the system would corrupt the responses of task-informative neurons. However, we suggest that this noisy task-irrelevant modulator plays a key role in solving the mystery of decoding. Specifically, we propose that the modulatory fluctuations serve as a label for the task-relevant neurons, helping the decoder to select these neurons for readout. Specifically, we posit that the decoder makes use of the modulator itself (or the modulator-induced covariability) when assigning appropriate decoding weights to each neuron. We construct such a *modulator-guided decoder*, and show through simulations that moderate levels of task-specific stochastic modulation of an encoding population can lead to a substantial overall benefit in decoding accuracy, while keeping the assumed knowledge about the encoding population at a biologically plausible level. Thus, structured noise may be an essential feature of brain computation, which could guide AI algorithms to overcome an essential gap to human behavioral performance.

## 2   Encoding/decoding models

To test our hypothesis, we simulate encoding in a population of stimulus-selective, noise-modulated Poisson neurons [13] and compare statistically optimal *ideal observer* decoders, that have full knowledge of the stimulus-selectivity and modulatory structure of the encoding population, with *biologically plausible* decoders, that must operate with limited knowledge of the encoding population.

**Encoding model: Poisson spiking population with task-targeted modulation**

The variability in spike count response $k_t$ over repeated presentations $t$ of a stimulus $s$ reflects the stochastic nature of neural spiking, commonly modeled using a Poisson point process with stimulus-dependent firing rate $\lambda(s)$. We account for supra-Poisson variability in neural responses, by introducing additional sources of stochasticity [11, 15, 16]. Specifically, the stimulus-driven rate of neuron $n$ is dynamically modulated by a time-varying signal $m_t$ [13], which leads to a doubly stochastic spiking process:

$$k_{nt}(s, m_t) \sim \text{Poiss}\left(\lambda_n(s)g(m_t)\right), \tag{1}$$

where $g(\cdot)$ is a positive-valued link function, here an exponential, to guarantee a positive firing rate.

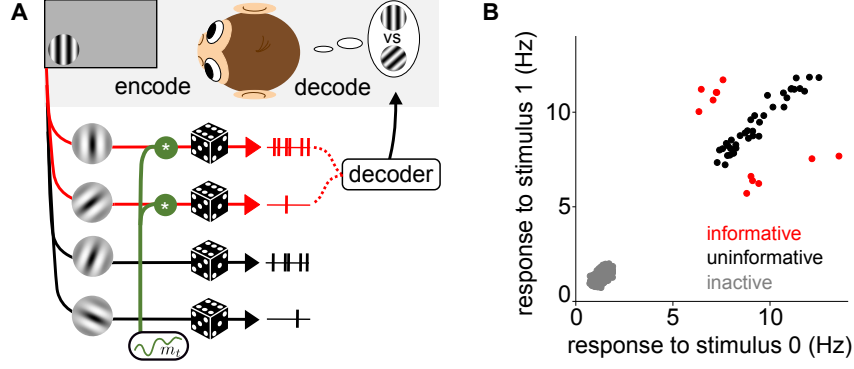

Figure 1: Encoding model. **A.** The encoding population consists of stimulus-tuned Poisson spiking neurons. Shared stochastic modulation (green) targets preferentially task-informative neurons and acts as a multiplicative gain. The decoder uses the modulatory signal to identify the task-informative neurons, and combines their responses to arrive at a decision. **B.** Stimulus selectivity of the population, qualitatively matched to experimental data. Neurons fall into three categories, based on their mean response to each of the two stimuli in the discrimination task. Neurons that respond differentially to the two stimuli are informative (red). Neurons with substantial but nearly equal responses to both stimuli are uninformative (black). The remaining neurons are inactive (and thus also uninformative), showing weak responses to both stimuli (gray).

We simulate a binary discrimination task (i.e., discriminate $s = 0$ from $s = 1$) similar to the change-detection task used in [17]. Empirical observations in macaque area V4 show that the modulatory signal $m_t$ is low-dimensional, shared across the neural population, and selectively targets neurons in proportion to their task-informativeness [13]. To capture these effects, we assume a one-dimensional modulator and introduce neuron-specific modulation weights, $w_n$, that are proportional to the $n$th neuron's ability to discriminate the two stimuli. Overall modulation strength in the population is determined by the modulator variance ($\text{var}(m_t w_n) = \sigma_m^2 w_n^2$ - see also [18]).

$$k_{nt}(s, m_t) \sim \text{Poiss}\left(\lambda_n(s) \exp(w_n m_t)\right). \tag{2}$$

Following a previous encoding model [13], we assume i.i.d. zero-mean Gaussian noise and variance $\sigma_m^2$ for $m_t$. Given the exponential nonlinearity, the modulatory factor causes an increase in spike count by $\exp\left(\frac{\sigma_m^2 w_n^2}{2}\right)$. To remove trivial benefits of the modulator due to an increase in firing rates, we correct for this expected increase by normalizing the firing rates in the encoding model:

$$k_{nt}(s, m_t) \sim \text{Poiss}\left(\lambda_n(s) \exp\left(w_n m_t - \frac{\sigma_m^2 w_n^2}{2}\right)\right). \tag{3}$$

**Statistically optimal "ideal observer" decoders**

Given the modulated Poisson encoding model, an ideal observer with complete knowledge of both stimulus response properties $\{\lambda_n(s)\}$ and modulation $\{w_n, m_t\}$ provides an upper-bound on task-decision accuracy. It operates by comparing the probability of the two stimuli under the full model (equivalently, by examining the sign of their log odds). For our modulated Poisson encoding model (see Eq. 3), this reduces to comparing a weighted linear combination of the observed neural spike counts against a time-varying threshold that is a function of the modulator (see derivation in Suppl. Info. S1). We refer to this as the *modulator-conditioned maximum likelihood* (MC-ML) decoder[1]:

$$\sum_n a_n^{(\text{MC})} k_{nt} > c_t^{(\text{MC})}, \tag{4}$$

with weights:

$$a_n^{(\text{MC})} = \log(\lambda_n(1)) - \log(\lambda_n(0)), \tag{5}$$

and *time-varying* threshold:

$$c_t^{(\text{MC})} = -\sum_n \exp(m_t w_n) \left[\lambda_n(1) - \lambda_n(0)\right], \tag{6}$$

where $\lambda_n(s)$ denotes the mean response of the $n$-th neuron to stimulus $s$ when $m_t = 0$.

The MC-ML decoder provides an upper bound on achievable performance, and relies on perfect knowledge of the modulator $m_t$, the stimulus selectivity of the neurons, $\lambda_n(s)$, and the coupling weights $w_n$. We can relax these requirements, by assuming that the modulator is unknown, and only the modulator-marginalized stimulus selectivity of the cells is available (i.e., the stimulus response averaged over possible modulators - see Suppl. Info. S1). We refer to this solution as the *modulator-marginalized maximum likelihood* (MM-ML) decoder. Due to the particularities of the Poisson noise model, this second decoder also computes a weighted sum over responses:

$$a_n^{(\mathrm{MM})} = \log(\lambda_n^*(1)) - \log(\lambda_n^*(0)). \tag{7}$$

But it compares this weighted sum to a *fixed* threshold:

$$c^{(\mathrm{MM})} = - \sum_n \left[ \lambda_n^*(1) - \lambda_n^*(0) \right], \tag{8}$$

where $\lambda_n^*(s)$ is the mean response of the $n$th neuron averaged (marginalized) over possible modulator values. For the encoding model in Eq. (3), $\lambda_n^*(s) = \lambda_n(s)$, which means that the decoding weights are the same as those used in the MC-ML decoder (i.e., $a_n^{(\mathrm{MM})} = a_n^{(MC)}$). Hence, in the case of a binary discrimination task, the MM-ML decoder is able to achieve an unbiased estimate of the decoding weights from the stimulus responses, without knowing the modulator. However, it does lead to systematic time-dependent biases in the decoder threshold and therefore to biased decisions.

**Biologically plausible decoders**

The MC-ML and MM-ML decoders are not plausible as a description of decoding in the brain, but they do provide a useful yardstick against which to compare the performance of more realistic decoders. They also motivate the use of a linear-threshold functional form for the solution. We now seek decoders of this form, that satisfy three criteria: (1) they are biologically plausible, in that they do not rely on detailed *knowledge* about the encoding population (neither the stimulus responses, nor the modulation weights), (2) they are behaviorally plausible, in that they have the ability to efficiently *adapt* to changes in task structure, so as to reflect the flexibility seen in monkey behavior [17, 9], and (3) they achieve *accuracy* approaching that of the optimal decoders.

We start with the simplest decoder, motivated by early work on neural binary discrimination/detection [1], assuming minimal knowledge of the encoding population, in line with our first criterion. The idea is to average the response of two sub-populations ("preferred" and "anti-preferred") and then compare these averages. Hence, the problem of learning decoding weights is reduced to choosing which population each neuron is assigned to; this is mathematically equivalent to determining the signs of a weight vector containing values $\pm 1$. For this reason, we refer to this model as the *sign-only* (SO) decoder. The signs are optimally estimated by comparing the mean responses to the two stimuli. This solution is agnostic to the details of the encoding model.

In order for this decoders to satisfy our second criterion – decoding flexibility – we need to estimate the signs given few trials. Indeed we see that classification into the two signed groups reaches high (90%) accuracy with only a few tens of trials, assuming low to moderate modulator strength (see Fig. 2A). If all neurons in a population were informative, learning the signs would provide an accurate readout of task information and the SO decoder would successfully fulfill also the last criterion (decoding accuracy). However, neural populations are diverse, and would generally be expected to include many uninformative neurons [19, 7]. The exact percentage depends on the neural population and behavioral task. We assess this parameter in detail in the section on decoder accuracy. An illustration of such an encoding population is given in Fig. 1 B which shows average responses of simulated neurons with diverse stimulus tuning features to two task-specific stimuli. Only a small fraction of neurons are responsive, while the large majority of neurons respond weakly ("inactive").

If the noise from these inactive neurons is not excluded by the decoder, it could still corrupt the signal [1]. We assessed decoding performance (% accurately discriminated stimuli) as a function of the number of inactive neurons (Fig. 2B). The SO decoder includes inactive neurons and assigns them to one decoding group or the other based on noise alone. Even though the individual noise of each inactive neuron is small by definition, together their task-irrelevant response eventually dominates the relevant stimulus signal (see Fig. 2B). In order to discount the inactive neurons, they should

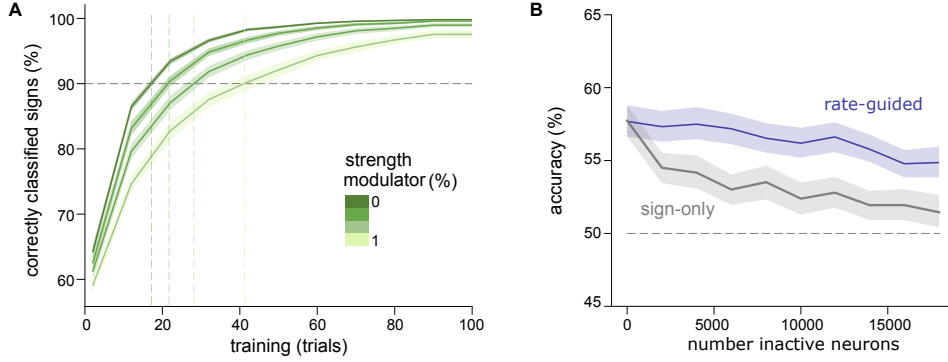

Figure 2: Accuracy of sign estimation, for simulated data. **A.** Mean % correctly attributed signs for informative neurons as a function of number of training trials with varying relative modulator strength (percentage of spike count variance of the informative neurons accounted for by the modulator). Decoding signs are learned within a few tens of trials. **B.** Mean performance of RG and SO decoders as the number of inactive neurons is increased. The RG decoder downweights inactive neurons, thus allowing it to maintain better performance than the SO decoder.

be assigned decoding weights with smaller amplitudes. The limited knowledge constraint, would however mean that these weights cannot be assumed to be known, but must be learned/adapted based on information readily available to upstream circuits.

Since informative neurons necessarily have to show activity during a task, one simple heuristic rule is to set decoding weights proportional to the mean spike count of their associated neurons:

$$|a_n^{(\text{RG})}| \propto \frac{1}{T}\sum_t k_{nt}. \tag{9}$$

For this decoder, the sign of the weights must again be learned (as for the SO decoder). The time-invariant threshold is set optimally. This *rate-guided* (RG) decoder improves decoding accuracy over the SO decoder by excluding neurons that do not respond to the stimuli (Fig. 1B, grey points). Fig. 2B shows that while the SO decoder's performance drops to chance level with increasing numbers of inactive neurons, the RG decoder is much less affected. However, the RG decoder is still far from optimal. In particular, it cannot exclude neurons that are active, but respond similarly to both stimuli (and are thus uninformative - Fig. 1B, black points).

The modulator could deliver this missing differentiation through its task-specific targeting structure. Here we propose a simple local rule for learning the modulation weights, by taking the inner product of spike counts and the modulator:

$$|a_n^{(\text{MG})}| = \frac{1}{T}\sum_t m_t k_{nt} \tag{10}$$

This *modulator guided* (MG) decoder satisfies the first criterion (biological plausibility), as it only assumes knowledge about the low-dimensional modulator, and the second (flexibility), as it learns absolute decoding weights on the fast modulator time scale, instead of the slow time scale dictated by the task-feedback arising from each trial.

Our heuristic learning rule results in estimates of the form (see Suppl. Info. S2):

$$\mathbb{E}\left[|a_n^{(\text{MG})}|\right] = \overline{\lambda}_n \sigma_m^2 w_n, \tag{11}$$

which scale with the average response of neuron $n$ across stimuli, $\overline{\lambda}_n$, and the modulator variance, $\sigma_m^2$. For this to be an unbiased estimate of the optimal decoding weights, we need the modulation strength to scale as $w_n = \overline{\lambda}_n^{-1}|a_n^{(\text{MC})}|$. This additional assumption for the encoding model will not affect the optimal decoding weights, but will change the expression for the optimal threshold (see Eq.6). We use this bias-corrected encoder here. Empirically, we have found that the positive effects of modulation on decoding remain, even in the absence of de-biasing.

Note that the above expression only provides the magnitude of the weights. The corresponding signs must be separately estimated, as for the SO decoder. For simplicity we assume that the MG threshold

| Decoder | Stimulus response knowledge | Modulation knowledge | Degrees of freedom |
|---------|------------------------------|----------------------|--------------------|
| MC-ML | $\lambda_n(s)$ (modulator-conditioned) | $m_t, w_n$ | 2N+N+T |
| MM-ML | $\lambda_n^*(s)$ (modulator-marginalized) | $\sigma_m, w_n$ | 2N+N+1 |
| MG | none | $m_t$ | T |
| RG | none | none | 0 |
| SO | none | none | 0 |

Table 1: Knowledge assumed by each of the five decoders (modulator conditioned: MC-ML, modulator marginalized: MM-ML, modulator guided: MG, rate guided: RG, sign only: SO - see text for details). Last column gives the dimensionality of variables that are assumed known or need to be estimated from neural responses, with $N$ the number of neurons in the population, and $T$ the number of time points.

has the optimal functional form, as defined by the MC-ML decoder (Eq.6). To maintain biological plausibility, we replace the true $w_n$ (which requires precise knowledge of the encoding model) with estimates $|\tilde{a}_n^{(\mathrm{MG})}|$. Furthermore, the difference in firing rates $[\lambda_n(1) - \lambda_n(0)]$ is replaced by an empirical estimate $\Delta\lambda$; this is determined as a function of the estimated decoding weights, the learned signs and one free parameter per informative subpopulation (two parameters in total). It measures the population average change in activity as a function of the stimulus and can easily be learned within a few trials.

## 3 Decoder accuracy

We tested the decoders listed in Table 1 in a binary discrimination task that evokes differential responses in a small subset of cells in the encoding population. We quantified decoding performance for discrimination between two stimuli $s = 0$ and $s = 1$, as we varied the overall strength of modulation. Results are shown in Fig. 3A. The MC-ML decoder provides a strict upper bound on decoding performance, as it assumes full knowledge of the encoding model. As the modulator variance $\sigma_m$ increases, the performance of this decoder monotonically decreases, confirming the intuition that the injecting correlated noise in task-relevant neurons is detrimental for encoding. For the encoding population tested here, the MM-ML decoder is nearly as good as the MC-ML decoder, however, performance falls faster with increasing modulator strength. This is due to the use of a fixed threshold, that does not adjust to temporal fluctuations of the modulator.

Among the biologically plausible decoders, SO performs near chance level, as it is unable to pick up the signal of the few informative neurons in the population. The RG decoder performs only slightly better, since it cannot differentiate between informative and uninformative neurons. Interestingly, the MG decoder performance shows a non-monotonic dependence on modulator strength. At low levels of modulation, performance increases with modulator strength - in this regime, the modulation allows the decoder to assign larger weights to the most informative cells. At higher levels, performance decreases, as with the ideal decoders, reflecting the corruption of the encoded signal. Hence, there exists an ideal range of modulation at which the MG decoder reaches its best performance, which is close to that of the ideal MC-ML decoder. Note also that since the MG decoder is capable of adjusting its threshold over time, depending on the modulator, it outperforms the MM-ML decoder (which uses a constant threshold) in the regime of high modulator strength.

We study this optimum with respect to modulator strength by looking at the encoding and decoding processes separately. For encoding, we predict the signal-to-noise ratio using Fisher's Linear Discriminant (FLD):

$$\mathrm{SNR} = \frac{\left(\mathbf{a}^T(\mu_1 - \mu_0)\right)^2}{\mathbf{a}^T\Sigma_1\mathbf{a} + \mathbf{a}^T\Sigma_0\mathbf{a}} \tag{12}$$

for the optimal decoding weights $\mathbf{a} = \mathbf{a}^{(MC)}$ (Fig. 3B, bottom). Decoding accuracy was estimated by the MSE of the MG-estimated decoding weights relative to the theoretical optimum. Given that the MG-decoding weights are unbiased, the MSE is given by the variance of the estimator, which decreases in inverse proportion to $T$ (see Suppl. Info. S2):

$$\mathrm{Var}\left[|a_n^{(\mathrm{MG})}|\right] = \frac{\sigma_m^2}{T}\left(\overline{\lambda_n}(1 + \sigma_m^2 w_n^2) + \overline{\lambda_n^2}e^{\sigma_m^2 w_n^2}(1 + 4\sigma_m^2 w_n^2) - \overline{\lambda}_n^2 w_n^2 \sigma_m^4\right), \tag{13}$$

where $\overline{\lambda}_n$ and $\overline{\lambda_n^2}$ are the mean and second moment of the neural response across stimuli.

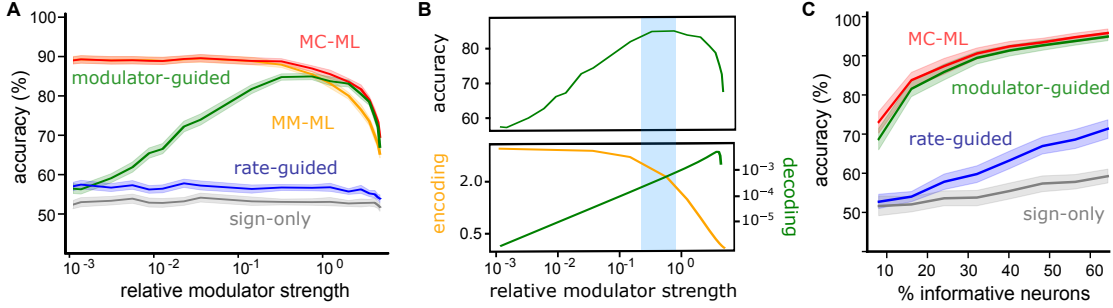

Figure 3: Comparison of different decoders, on simulated data. **A.** Decoding accuracy as a function of relative modulator strength (percentage of total spike count variance that can be attributed to the modulator in informative neurons). Lines indicate mean accuracy (% correct), and shaded region its $95\%$ confidence interval. We simulated 5000 cells in total, of which 50 were active cells and of those 12 ($24\%$ of active) were informative cells. Baseline firing rates were set similar for all active neurons. **B.** Increasing modulator strength has opposite effects on encoding and decoding accuracy. It decreases the FLD ratio (encoding accuracy), but it also increases decoding accuracy, measured as the inverse of the MSE; these two effects jointly produce the maximum in accuracy of the MG-decoder (blue shaded region). **C.** Performance of different decoders as a function of the number of informative neurons (in % informative neurons of active neurons). The strength of modulation was set fixed to the MG-optimal strength (see A). Other parameters are the same as in A.

We also tested the influence of percentage of informative neurons in the encoding population on these results. The decoding problem of identifying task-informative neurons is particularly difficult when only very few of the active neurons are task-informative. In experiments, the percentage of informative neurons varies depending on the intrinsic tuning properties of the cells (e.g. width of tuning curves), and extrinsic task properties (e.g. coarse vs. fine discrimination). In our simulations, varying the percentage of informative neurons serves as a proxy for both. Unsurprisingly, increasing the percentage of informative neurons increases decoding accuracy across the board (Fig.3C). While the SO decoder improvements are modest, the RG decoder achieves reasonable accuracy if more than half of the neurons are informative. The advantage of the MG decoder over RG is strongest if the fraction of informative neurons is small. These results are robust to changes in the overall size of the population (not shown). Overall, this suggests that, under realistic conditions, our proposed modulator-guided decoding mechanism provides a substantial benefit over simpler solutions.

Finally, the modulation strengths above were set to the optimal decoding weights for every neuron, hence assuming high precision targeting and no additional sources of noise (Eq.3). In Fig.4 we show numerically that our results are robust to noise in the modulator weights, as well as to perturbations in the firing rates of the neurons, in the form of additive Gaussian noise.

## 4 Discussion

Artificial neural networks may excel at solving the one task they have been trained for, but require substantial retraining when goals change. In contrast, human and animals can rapidly and flexibly switch between goals, with existing neural resources quickly recruited for the task at hand. Here we proposed that a functionally targeted stochastic modulator [13] could dynamically label informative neurons, facilitating their flexible and accurate task-specific readout. We showed that a modulator-guided linear decoder, in which weights are estimated through correlation of responses with the modulator, can achieve near-optimal performance. We investigated how parameters of the encoder (proportion of inactive neurons, and active but uninformative neurons) impact performance and found that these dictate a choice of modulator strength that best balances the disruptive effects of correlated noise on encoding against its positive effects for decoding. Importantly, performance is invariant to other parameter changes, such as size of the population and baseline firing rate, demonstrating the robustness of the modulation labeling scheme to circuit details.

Historically, ideal observer models have ignored the presence of modulation, yet have provided good approximations of behavioral performance. Our MM-ML ideal observer provides a possible explanation for this incongruity: an experimenter that measures tuning functions by averaging neural responses in the presence of unaccounted-for modulation is effectively marginalizing over

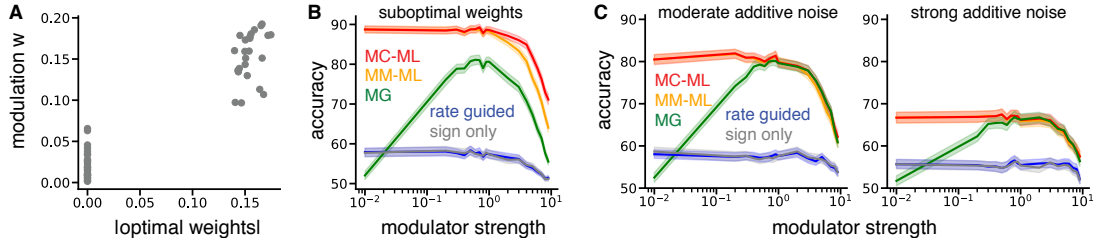

Figure 4: Robustness of model to perturbations in modulation weights and firing rates. **A.** Modulation strengths $w_n$ set to the absolute value of optimal decoding weights corrupted by additive independent Gaussian noise. **B.** Decoder performance with noisy modulation weights. **C.** Decoder performance with moderate and high levels of Gaussian noise added to the firing rates defined by Eq.3.

it. Optimal decoding weights derived from these estimates are in fact correct, but the use of a fixed decision threshold is suboptimal. This suboptimality is relatively minor in the context of our simulations (compare MC-ML and MM-ML in Fig. 3A), but could prove more substantial when fit to physiological data. Another source of bias, often ignored in ideal observer analyses, arises from the selection of experimentally recorded neurons. Neural recordings are generally biased towards active neurons, partly because low firing neurons are more likely to be overlooked, and partly because experimental stimuli are often optimized to drive the recorded population. While the signal may be concentrated in the recorded subpopulation, a downstream decoding area must also process the substantially larger unrecorded population.

Our encoding model assumes multiplicative noise since, to date, there is no evidence that additive noise is functionally targeted. MOreover, experimental reports are conflicting as to whether additive noise is a common phenomenon (e.g. Goris et al. [11] argue that an additive noise model is inconsistent with their data). Should it be there, task-invariant additive noise would decrease the performance of all decoders, but would not qualitatively change our results (see Fig. 4 C). For simplicity, we have assumed a single task-specific signal that underlies the correlated noise within the population. This is consistent with [13], which showed that V4 noise correlations were largely captured using a one dimensional modulator per hemisphere. Alternatively, one could introduce several Gaussian modulators, that combine linearly to jointly gate neural responses. This model would be harder to parameterize, but the net effect would be similar. Additional modulators that are not targeted would reduce the SNR of all neurons and negatively affect all decoders, but again, should not qualitatively change the results.

Our theory does not specify the biological implementation of the modulation, in terms of where it is generated, the mechanism by which it targets encoding neurons in a task-specific manner, or the means by which it is made available to downstream circuits. In principle, dynamic changes in population noise correlations could arise through either local [20] or top-down mechanisms [14]. For the orientation discrimination task considered here, one could imagine taking advantage of the topological organization of the sensory code for orientation, modulating spatially-localized clusters of neurons in V1 without requiring explicit knowledge of their individual tuning. The induced noise correlations would then propagate bottom-up to higher areas, labeling task-relevant neurons that need not be topographically clustered. This mechanism predicts a hierarchy of modulator-based labeling across the sensory processing stages, consistent with the experimental observation that correlations between V1 and MT increase with behavioral performance, triggered by attending towards a stimulus [21]. Alternatively, on a slower time scale, the top-down connections could self-organize to allow for feature-selective noise targeting, akin to attentive mechanisms recently introduced in artificial neural networks [22, 23].

The topic of flexible information routing in the brain has a long history. In particular, the signal transmitted by sensory neurons is enhanced when their firing is synchronized, and thus, oscillations have been hypothesized to serve as labeling mechanism [24]. The "communication through coherence"(CTC) theory [25] has refined this idea in an encoding-decoding framework, where a top-down oscillatory signal projects to both encoding neurons with the same feature selectivity, and to the decoding network that reads out from them. Oscillations can play a similar role to our modulator, with some important distinctions. First, CTC considers a fixed labeling strategy, with oscillations targeting feature-selective neurons, while our framework focuses on the flexible learning through

targeting of task-informative neurons. The two proposals might be hard to distinguish in a detection task, but make distinct predictions for discrimination. Second, CTC assumes a decoder with fixed threshold, which (at least within our modeling framework) is suboptimal. Third, the two theories differ in the statistics of the labeling signal: CTC assumes periodic signals, while our model uses stochastic signals, assuming only a timescale. We note, though, that our framework could be readily adapted to the case of an oscillatory modulator.

Our model makes several predictions which can be examined in an experimental context that includes a dynamically changing task. In particular, the influence of low-dimensional (shared) noise should shift with the task, so as to continue to preferentially target task-informative neurons. Moreover, a modulator-guided decoder should outperform simpler strategies (sign-only or rate-guided) when applied to physiological data. Since our theory posits an optimal level of modulation relative to the stimulus-induced variance, we expect that attention shifts the level of modulation towards this (empirically estimable) optimum. Furthermore, the direction of the shift may provide clues regarding the mechanisms underlying noise generation [20].

Our decoders are designed for a classification (as opposed to estimation) task because the experiments showing task-specific modulation have been done with binary discrimination tasks. In principle, it might be possible to extend this framework to estimation, which also entails learning to appropriately weight informative neurons while ignoring uninformative ones. Modulator-labeling should prove useful in this context, although the details of the decoder will likely change. There is growing interest in the machine learning community in developing more flexible, adaptive neural models. Despite some recent progress, the design of artificial neural networks that can handle many tasks and leverage past learning to generalize to new tasks (e.g., multi-task, meta-learning, transfer learning) is in its infancy. Our proposal provides a novel biologically-inspired solution, as a potential step toward this goal.

**Acknowledgments**

This work was supported by the Google PhD Fellowship (CH).

## Footnotes

[1]For brevity, 'decoder' refers to both the stimulus readout, and its corresponding optimal discriminator.

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
