[Supplementary Material]

# Flexible information routing in neural populations through stochastic comodulation – Supplementary material –

**Caroline Haimerl**
Center for Neural Science
New York University
ch2880@nyu.edu

**Cristina Savin**
Center for Neural Science
Center for Data Science
New York University
csavin@nyu.edu

**Eero P. Simoncelli**
Center for Neural Science, and
Howard Hughes Medical Institute
New York University
eero.simoncelli@nyu.edu

## S1  Derivation of optimal decoders

Given the modulated Poisson model, $k_{nt}(s, m_t) \sim \text{Poiss}\left(\lambda_n(s) \exp\left(w_n m_t - \frac{\sigma_m^2 w_n^2}{2}\right)\right)$, and assuming that the modulator $m_t$ and the modulation weights $w_n$ are known, the log probability of the stimulus $s$ at time point $t$ given spike counts $k_{nt}$ of the whole population, $n = \{1, 2 \ldots N\}$, becomes:

$$\text{L}(s) = \sum_n^N k_{nt}\left(\log\lambda_n(s) + w_n m_t - \frac{\sigma_m^2 w_n^2}{2}\right) - \sum_n^N \lambda_n(s)\exp\left(w_n m_t - \frac{\sigma_m^2 w_n^2}{2}\right). \quad (1)$$

When discriminating between two stimuli $s = \{0, 1\}$ under this model, the optimal decision is given by the sign of log-odds ratio, $\text{L}(s = 1) - \text{L}(s = 0) \gtrless 0$, which translates into the following expression:

$$\text{L}(0) - \text{L}(1) = \sum_n^N k_{nt}\log\left(\frac{\lambda_n(0)}{\lambda_n(1)}\right) - \sum_n^N \exp\left(w_n m_t - \frac{\sigma_m^2 w_n^2}{2}\right)(\lambda_n(0) - \lambda_n(1)). \quad (2)$$

This corresponds to thresholding a weighted combination of individual neural responses, with optimal decoding weights:

$$a_n^{(\text{MC})} = \log\left(\frac{\lambda_n(0)}{\lambda_n(1)}\right). \quad (3)$$

Since $m$ is a known constant, it does not influence the decoding weights themselves, but merely changes the threshold. For the same reason, the optimal decoding weights remain unchanged whether the modulator is known (MC-ML), or whether its effects are marginalized over (MM-ML).

## S2  Properties of the modulator-guided decoder

For the encoding model described above, we aim to estimate decoding weights $a_n^{(\text{MG})}$ by correlating the response of neuron $n$ with the modulator $m$:

$$|a_n^{(\text{MG})}| = \frac{1}{T}\sum_t^T m_t k_{nt}. \quad (4)$$

Here we analyse the properties of this estimate. First, its mean is:

$$\mathbb{E}\left[|a_n^{(\mathrm{MG})}|\right]_{\mathrm{P}(k,s,m)} = \mathbb{E}\left[mk_n\right]_{\mathrm{P}(k,s,m)} \tag{5}$$

$$= \mathbb{E}\left[m\lambda_n(s)\mathrm{e}^{mw_n-\frac{\sigma_m^2 w_n^2}{2}}\right]_{\mathrm{P}(s,m)} \tag{6}$$

$$= \mathbb{E}\left[\lambda(s)\right]_{\mathrm{P}(s)} \mathbb{E}\left[m\mathrm{e}^{mw-\frac{\sigma_m^2 w_n^2}{2}}\right]_{\mathrm{P}(m)}, \tag{7}$$

$$= \bar{\lambda}_n \int m\mathrm{e}^{mw_n-\frac{\sigma_m^2 w_n^2}{2}}\mathrm{e}^{-\frac{m^2}{2\sigma_m^2}}\,dm \tag{8}$$

$$= \bar{\lambda}_n \sigma_m^2 w_n, \tag{9}$$

where $\bar{\lambda}_n$ denotes the average activation of the neuron, $\bar{\lambda}_n = \sum_s \mathrm{P}(s)\lambda_n(s)$; we have used the encoding model and the fact that $s$ and $m$ are independent (Eq. 7) and $m_t$ is i.i.d. gaussian with zero mean and variance $\sigma_m^2$ (Eq. 8). Under the assumption that $w_n = \log\frac{\lambda_n(1)}{\lambda_n(0)}$, the MG estimates of the decoding weights are biased. While the scaling with $\sigma_m^2$ could be easily corrected for by appropriately rescaling the threshold, the neuron-specific $\bar{\lambda}_n$ bias is problematic. One could correct this bias by a slight adjustment of the encoding model, i.e. assuming $w_n = \frac{1}{\bar{\lambda}_n}\log\frac{\lambda_n(1)}{\lambda_n(0)}$. This will not change the optimal decoding weights $a^{(\mathrm{MC})}$, but will affect the expression of the optimal threshold.

The variance of the estimator can be computed in a similar way:

$$\mathrm{Var}\left[|a_n^{(\mathrm{MG})}|\right] = \frac{1}{T}\left(\mathbb{E}\left[m^2 k_n^2\right] - \mathbb{E}\left[mk_n\right]^2\right) \tag{10}$$

$$= \frac{1}{T}\left(\mathbb{E}\left[m^2 k_n^2\right] - \left(\bar{\lambda}_n \sigma_m^2 w_n\right)^2\right) \tag{11}$$

The second moment term can be computed as:

$$\mathbb{E}\left[m^2 k_n^2\right]_{\mathrm{P}(k,s,m)} = \mathbb{E}\left[m^2\left(\lambda_n(s)\mathrm{e}^{mw_n-\frac{\sigma_m^2 w_n^2}{2}} + \left(\lambda_n(s)\mathrm{e}^{mw_n-\frac{\sigma_m^2 w_n^2}{2}}\right)^2\right)\right]_{\mathrm{P}(s,m)}$$

$$= \bar{\lambda}_n \int m^2 \mathrm{e}^{mw_n-\frac{\sigma_m^2 w_n^2}{2}}\mathrm{e}^{-\frac{m^2}{2\sigma_m^2}}\,dm + \overline{\lambda_n^2}\int m^2 \mathrm{e}^{2mw_n-\sigma_m^2 w_n^2-\frac{m^2}{2\sigma_m^2}}\,dm$$

$$= \bar{\lambda}_n \int m^2 \mathrm{e}^{-\frac{(m-\sigma_m^2 w_n)^2}{2\sigma_m^2}}\,dm + \overline{\lambda_n^2}\mathrm{e}^{\sigma_m^2 w_n^2}\int m^2 \mathrm{e}^{-\frac{(m-2\sigma_m^2 w_n)^2}{2\sigma_m^2}}\,dm$$

$$= \bar{\lambda}_n(\sigma_m^2 + \sigma_m^4 w_n^2) + \overline{\lambda_n^2}\mathrm{e}^{\sigma_m^2 w_n^2}(\sigma_m^2 + 4\sigma_m^4 w_n^2)$$

where $\overline{\lambda_n^2} = \sum_s \lambda_n^2(s)\mathrm{P}(s)$ denotes the second moment of $\lambda_n(s)$ and we have used the fact that the second moment of a Poisson distribution with mean $\lambda$ is $\lambda + \lambda^2$, the fact that each of the two integrals is the second moment of a gaussian. This holds for any setting of $w_n$ (with or without unbiasing).

Lastly, the covariance for the decoding weights of pairs of neurons $n, l$ takes the form:

$$\mathrm{Cov}\left[|a_n^{(\mathrm{MG})}|, |a_l^{(\mathrm{MG})}|\right] = \frac{1}{T}\left(\mathbb{E}\left[m^2 k_n k_l\right] - \mathbb{E}\left[mk_n\right]\mathbb{E}\left[mk_l\right]\right) \tag{12}$$

$$= \frac{1}{T}\left(\bar{\lambda}_{nl}\mathbb{E}\left[m^2 \mathrm{e}^{m(w_n+w_l)-\frac{\sigma_m^2(w_n^2+w_l^2)}{2}}\right] - \bar{\lambda}_n\bar{\lambda}_l\sigma_m^4 w_n w_l\right) \tag{13}$$

$$= \frac{1}{T}\left(\bar{\lambda}_{nl}\mathrm{e}^{\sigma_m^2 w_n w_l}\left(\sigma_m^2 + \sigma_m^4(w_n+w_l)^2\right) - \bar{\lambda}_n\bar{\lambda}_l\sigma_m^4 w_n w_l\right) \tag{14}$$

where $\bar{\lambda}_{nl} = \sum_s \lambda_n(s)\lambda_l(s)\mathrm{P}(s)$ is related to the signal correlations of the two neurons.

## S3  Population size: number of inactive neurons

In Figure 2B of the main text we simulated an increasing encoding population size by adding inactive neurons[1] and demonstrating the different effect on performance of the sign-only decoder and the

**Figure S1:** Performance of all decoders as more inactive neurons are added to the encoding population. The number of active neurons is fixed at 50 and the number of informative neurons at 12.

rate-guided decoder. Here we repeat this simulation using all decoders. The optimal decoders (MC-ML and MM-ML) are unaffected by the increase in inactive neurons, while the MG decoder, as the RG decoder, shows a slight decrease in performance due to the added noise. Neither is affected to the same extent as the SO decoder, which drops to chance. This reflects the fact that SO cannot average out the noise from the inactive neurons).

## Footnotes

[1]Inactive neurons are task-irrelevant, and have low but nonzero firing rate.