[Reviews · NeurIPS 2019]

Reviewer 1



The authors propose a neurally plausible model for online recognition of task-relevant neurons. The key implication is that stochastic shared modulation acts as a “tag” for task-relevant neurons so that the downstream population can quickly and easily learn the important neurons for subsequent computations. As a proof-of-concept the authors apply this theory to the construction of a decoder for a simulated binary classification task but the theory plausibly extends beyond this regime. The biggest caveat to the theory is that it is unclear how the task-relevant neurons would be tagged in the first place, but the authors are up-front about this in the discussion. Overall, the paper is exceptionally well written and is well motivated with relevant literature in the Introduction. The mathematics were easy to follow as was the reasoning for each choice of decoder classification rules. While most of the figures were well-selected and complimented the ideas in the paper effectively, Figure 3B is difficult to parse. The figure is not well labeled and the color choices appear to obscure the plotted curves on one side or the other of the vertical line. Does this line correspond to the optimum from panel A? Which decoder/model is this in reference to? Is there something I’m missing about why it is plotted this way? The authors used the MSE to asses decoding performance. Other than the fact that MSE decomposes into bias and variance, would it not have been a more natural choice to use the cross-entropy?

Reviewer 2



I’d like start by saying that I enjoyed reading this manuscript. I found that this submission contains several interesting ideas. The theoretical proposal about the function of gain modulation seems to be novelty, although it heavily relies on a specific model that has been proposed previously (Rabinowitz, et al., 2015). The writing is generally clear and the structure of the paper is well organized. My major concern is that based on the current version, it is unclear how robust and general these results are (detailed below). While the set of results presented are interesting, they are rather limited in scope. I feel it is a bit difficult to judge the significance of the contributions. 
- the paper only considered a binary classification task. Would the results generalize to other tasks such as estimation tasks (which is widely used in psychophysics and neurophysiology, and also ecologically important)? - the model assumes a one-dimensional shared modulator. Realistically the shared modulation of neural responses in the brain might be multi-dimensional. In fact, the results in [Rabinowitz, et al., 2015] suggest more than 1 modulator. Could the model be generalized to deal with multiple modulators? - the current results critically rely on the assumption of pure multiplicative gain. However, recent experimental results [ e.g., Lin et al., (2015), Aranda-Romero et al.. (2016)] suggest that there is also an additive component in the fluctuation of V1 neurons. Is the proposed model flexible enough to deal with that? - there are multiple hyper-parameters in the model, such as proportion of activity cells and proportion of informative neurons. Are the results presented in Fig 3 robust to the choice of these parameters? Another main concern I have is- what concrete experimental predictions does this model provide in order to validate/falsify it? Also, is it possible to incorporate the prior into the framework? Minor points: line 167-168: there is the claim about 90% accuracy. It is a bit mysterious to me where this number comes from. Presumably the performance of the decoders would depend on other parameters such as the number of neurons. It seems premature and a bit misleading to claim RG decoder doesn’t achieve a behavioral level of performance. Fig 3B is a bit confusing. Would it be helpful to leave some gap between the different blocks? Reference: Arandia-Romero, Iñigo, et al. "Multiplicative and additive modulation of neuronal tuning with population activity affects encoded information." Neuron 89.6 (2016): 1305-1316. Lin, I-Chun, et al. "The nature of shared cortical variability." Neuron 87.3 (2015): 644-656. Rabinowitz, Neil C., et al. "Attention stabilizes the shared gain of V4 populations." Elife 4 (2015): e08998. *************after the rebuttal I appreciate the authors' response. The proposed discussion on the role of additive noise would make the paper stronger. It would also be useful to have an explicitly discussion about the different tasks( e.g. estimations task) and the how the prior information would fit into the current framework. This would make it a more balance paper with more careful treatment of the potential pitfalls. I have increased score from 6 to 7 to reflect these points.

Reviewer 3



The authors propose a model of encoding information across a neural population for a discrimination task in which the firing rate of neurons across a population are modulated by a shared low-dimensional gain signal, where neurons which are more informative for a task are more strongly coupled to the gain signal. The authors motivate their work by experimental findings of such gain signals and couplings. The authors propose a decocer which leverages these modulatory signals to identify task-informative neurons. They compare this decoder to ideal-decoders, informed by knowledge of the encoding model, as well as simpler (more biologically plausible decoders) which do not leverage the modulatory signals. They simulate a binary discrimination task and show the performance of the simpler decoders, ingorent of the modulatory signal, decreases as the strength of a task-independent modulatory signals increases and as more neurons without strong task tuning are added to a population. They also test the performance of their more biologically plausible decoder which does leverage the modulatory signal and find it achieves performance comparable to the ideal decoders. I provide my detailed comments below, but think the overall idea is likely to be of interest to the community, while the simulations could be extended to strengthen the claims of biological plausibility and it would help to clarify how the encoding/decoding scheme presented here fits in with the larger picture of how noise correlations/modulatory signals change with attention. Originality: To my knowledge, the proposed decoder, leveraging a low-dimensional, task-independent moulatory signal shared across a population is novel. I find this idea intriguing and think others in the community would find it of interest as well. Quality: The quality of the paper is fairly high. In some areas, the clarity of the exposition could be improved, which I have noted in the clarity section below. With that said, there are at least ways the quality of the results could be improved. First, the proposed encoding scheme seems to rely on knowledge of the optimal decoding weights for each neuron (the coupling of each neuron to the modulatory signal is proportional to its decoding weight). This seems like a strong assumption. Would the encoding/decoding scheme still work if they were weakly (at least not perfectly) correlated? Second, the decoding scheme relies on learning the signs of weights from repeated trials. The authors claim this can happen in a “handful” of trials - while I have no reason to doubt them - demonstration of this would be helpful. As a large part of the motivation for the work is quickly identifying informative neurons to respond quickly to changing task demands, explicit results showing the benefit of the encoding/decoding scheme with regards to this would be helpful. Clarity: The paper is fairly clear. However, description of some of the mathematical equations could be improved. For example, on page 2 k_t is first introduced by it is not clear until later that t indexes time. Similarly when first presented with equation 1, it is not clear what subscript n is, and why it does not appear on the left hand side of that equation (which I believe is a typo)? Attention to details like this throughout the paper would aid reader’s understanding. Significance: I find the motivation for the proposed decoder, that of leveraging a task-independent modulatory signal to identify task-informative neurons, very interesting. However, I am struggling to determine how significant this model may be in the context of the results the authors themselves cite as motivation of the work. In particular, the author’s cite Rabinowitz et al. 2015. That work analyzes simultaneous recordings from both hemispheres of V4 and find that the variance of modulatory signals in a set of neurons with receptive fields in an attended part of space decreases. The authors make no mention of this finding but instead cite Rabinowitz et al. for another finding - that the coupling of a neuron to modulatory strength is correlated to it’s informativeness for a task. Would the encoder/decoder model proposed in the present work still suffice to identify the task informative neurons in this scenario - where the variance of the modulatory signal in the neurons more informative for a task (in the attended part of space) decreases? Specifically, would the decoder ignore neurons in the unattended part of space in favor for neurons in the attended part of space, even through the variance of the modulatory signal for neurons in the attended part of space decreases? Clarity on this question would improve understanding of the significance of the work. With that said, even if the encoding/decoding scheme proposed here is only one of a number of ways the brain identifies task informative neurons, I find the ideas in the paper to be interesting. Update after author response: I thank the author's for their response. It is reassuring to know the theory is robust to non-ideal weights. I also appreciate the clarification that the role of attention in reducing the variance of the modulatory may be to get the modulatory signals into some ideal range. I feel my main concerns with the paper have been addressed, and I have increased my score to reflect this.

[Author Response · NeurIPS 2019]

We thank the reviewers (R1, R2, R3) for the helpful comments, corrections and suggestions.

The main concern seems to be the **robustness** of our results to various deviations from the idealized scenario considered
by our theory (R2, R3). First, while the **hyperparameter** space is too large to explore systematically, simulations
suggest the qualitative phenomenology presented in the paper is robust to various model details. The effects of varying
the fraction of active/informative neurons are shown in Fig. 2B/3C (we will improve 2B to include all decoders
(R2)). We will also document additional parameter variations in the supplementary material (SM). Second, R2 and R3
expressed concern about the precision required for the **optimal modulation weights** in the encoding. Here we show
numerically that the results hold qualitatively even when the modulation deviates significantly from the absolute optimal
decoding weights ($w = dec_{ML} + \epsilon$ where $\epsilon$ is independent gaussian noise; example in panel A) although the overall
performance degrades (B). Hence our idea could still apply to a more realistic suboptimal encoding model.

R2 also identified several potential mismatches between our idealized model and real data. First, the modulator could
be **multidimensional**. We chose to focus on the unidimensional case because: 1) it is the simplest, 2) mathematically,
having a linear combination of gaussian, task-specific modulators does not qualitatively change the problem, though
it makes the model harder to parametrize (additionally, if some modulator targeting is not task specific, that would
reduce the SNR of all neurons and correspondingly affect all decoders), and 3) while in the Rabinowitz paper there
were several modulators, most of variance was accounted for by one (for each hemisphere). Second, the presence of
**additive noise**: experimental reports are conflicting (e.g. Goris et al.2014 argue that additive noise is inconsistent with
their data); moreover, to date, there is no evidence that this component is functionally targeted. Simulations show that
task-invariant additive noise decreases the performance of all decoders, but does not qualitatively change our results
(panel C, to be included in SM). These issues will be further detailed in the Discussion.

We chose to focus on classification rather than **estimation** because the experiments showing task-specific modulation
use binary discrimination. As R2 rightfully points out, it is important to expand the theory to other tasks. In principle,
since estimation also entails learning to appropriately weight informative neurons while ignoring uninformative ones,
modulator-labeling should be helpful there too, though the details of the best encoder and decoder will likely change.
Including an informative **prior** (R2) should not qualitative change the discussion (it only shifts the threshold), unless the
prior directly affects the pattern of modulation. To our knowledge, there is no experimental data supporting this idea.

The model makes several **experimental predictions** (R2), which we are in the process of testing using V1 monkey
recordings in a task that dynamically shifts the task-relevant sub-population. The main assumptions of the theory
to check: 1) the subpopulation of task-informative neurons is small and hence hard to identify, 2) low-dimensional,
shared noise, changing faster than the trial duration, that preferentially targets informative neurons. We further predict
that our modulator-guided heuristic decoder should outperform simpler strategies (sign-only or rate-guided). Perhaps
counterintuitively, **attention** reduces the variance of the modulator (Rabinowitz et al). Since our theory posits an
optimal level of modulation relative to the stimulus-induced variance, we would suggest that attention shifts the level
of modulation towards this (empirically estimatable) optimum. The direction of the shift may provide hints about the
mechanics of noise generation, something we know little about at the moment (Huang et al. 2019).

There is a small misunderstanding regarding the usage of **MSE** (R1), which we use only for the *modulation weights*
estimator, but not for assessing task performance (measured as % correct). Similarly, the procedure used for the **learning**
of the MG decoding weights seems unclear (R2). It includes two processes: 1) learning the signs, which happens using
end-of-trial feedback, but only needs few examples (see Fig.2A), and 2) learning the absolute optimal decoding weights,
which happens within the trial by estimating modulation weights, with variance scaling as $1/T$; the exact learning
rate is determined by the strength and time constant of the modulator (see Eq.14). Quantitatively confirming if the
modulation is enough to explain the animal's **speed of learning** requires further data analysis (ongoing).

Minor: the explicit reference of the target behavioral 90% level is somewhat misleading and will be removed (R2). We
will add the additional references (R2), clarify the equations (R2, R3) and improve Fig.3B visually and for clarification
(vertical line=optimum from 3A, middle panel should show precision instead of variance, to reduce confusion) (R1, R2).
All the above points will be incorporated in the camera-ready version of the paper, should our submission be accepted.



[Meta-Review · NeurIPS 2019]

This paper presents a model of sensory encoding that utilizes task irrelevant noise to improve decoding by a downstream model. Specifically, it is shown that if task-informative neurons are co-modulated by a low-dimensional, task-irrelevant noise signal, then a linear decoder that uses readout weights based on the modulation strength can achieve near-optimal accuracy. The reviewers agreed that this paper is a worthwhile contribution and interesting. There were some initial concerns/questions on a variety of topics (the use of multiplicative vs. additive noise, the robustness for different weights, etc.), but the author responses satisfied the reviewers and it was agreed that the paper should be accepted.